# Paroxetine treatment in an animal model of depression improves sperm quality

Reyhane Aghajani[1,2], Marziyeh Tavalaee[1,2], Niloofar Sadeghi[2], Mazdak Razi[3], Parviz Gharagozloo[4], Maryam Arbabian[2], Joël R. Drevet[5]*, Mohammad Hossein Nasr-Esfahani [1,2]*

1 ACECR Institute of higher Education (Isfahan branch), Isfahan, Iran, 2 Department of Animal Biotechnology, Reproductive Biomedicine Research Center, Royan Institute for Biotechnology, ACECR, Isfahan, Iran, 3 Faculty of Veterinary Medicine, Department of Histology, Urmia University, Urmia, Iran, 4 Celloxess LLC, Ewing, NJ, United States of America, 5 Faculty of Medicine, GReD laboratory, CNRS UMR6293-INSERM U1103-Université Clermont Auvergne, Clermont-Ferrand, France

* mh.nasr-esfahani@royaninstitute.org (MHN-E); joel.drevet@uca.fr (JRD)

Data Availability Statement: All relevant data are within the paper and its Supporting Information files.

## Abstract

Depression in mammals is known to be associated with poor reproductive capacity. In males, it has been associated with decreased efficiency of spermatogenesis as well as the production of spermatozoa of reduced structural and functional integrity. Although antide-pressants are effective in correcting depressive states, there is controversy regarding their effectiveness in restoring male reproductive function. Here, using an animal model of depression induced by a forced swim test, we confirmed that depression is accompanied by impaired male reproductive function. We further show that administration of a conventional antidepressant of the serotonin reuptake inhibitor class (paroxetine) impairs male reproductive performance in terms of sperm production and quality when administered to healthy animals. Intriguingly, when paroxetine is administered to "depressed" animals, it resulted in a complete restoration of the animal's ability to produce sperm that appears to be as capable of meeting the parameters evaluated here as those of control animals. The one-carbon cycle (1CC) is one of the most important metabolic cycles that include the methionine and folate cycles and plays a major role in DNA synthesis, amino acids, and also the production of antioxidants. Our results show that depression affects the main components of this cycle and paroxetine on healthy mice increases homocysteine levels, decreases glycine and vitamin B12, while in depressed mice, it increases folate levels and decreases vitamin B12. Thus, paroxetine exerts negative impacts on male reproductive function when administered to healthy animals and it well correlate with the altered sperm parameters and functions of depressed animals, and its mechanism remains to be explored.

## Introduction

Depression, is a complex disease with multiple genetic and epigenetic origins [1]. It is the second most common human pathology [2]. With the pandemic crisis of COVID-19, this situation is likely to worsen and depressive states are expected to become increasingly acute.

**Funding:** The author(s) received no specific funding for this work.

**Competing interests:** The authors have declared that no competing interests exist.

Depression is associated with an imbalance of brain chemicals called neurotransmitters, including serotonin, dopamine, norepinephrine, acetylcholine, GABA and glutamate, which regulate many physiological functions, including reproduction. Selective serotonin reuptake inhibitors (SSRIs) is a class of drugs, commonly used as antidepressants to treat depression and a range of associated disorders such as anxiety, post-traumatic stress disorder, social phobia, and obsessive-compulsive behaviors [3,4]. Several studies have shown that antidepressants could interfere with the hypothalamic-pituitary-gonadal (HPG) axis, block gonadotropin release from the pituitary gland, reduce testosterone secretion, and consequently disrupt spermatogenesis [5,6]. In support of these effects, low testosterone concentration has been observed in serotonin-treated rats associated with reduced sperm concentration, motility and normal morphology [7,8]. This may be explained by the fact that serotonin receptors have been identified in the mammalian testis and epididymis where they regulate testicular blood flow and sperm maturation [9]. Bhongade MB et al also demonstrated that the level of serum testosterone and the mean of sperm quality were lower in individuals having abnormal HADS (Hospital Anxiety and Depression Score) compared to patients having normal HADS [10]. In addition, Safarinejad showed that the mean of sperm DNA damage and sperm parameters were significantly lower in depressed men treated with SSRIs compared to fertile men [11]. Beeder et al also reported that selective serotonin reuptake inhibitors could negatively impact sperm quality *in vitro*, both in animal models and human studies (9). Other side effects of antidepressants could be associated to libido, delayed ejaculation or even impotence [9,12,13]. These off-target negative effects can lead to infertility, which in a vicious cycle is known to increase depressive and anxiety symptoms [14].

Previous studies also demonstrated that SSRIs such as paroxetine, sertraline and citalopram can interact with sperm plasma and mitochondrial inner membrane sulfhydryl groups and cause deleterious effects on motility, viability, capacitation and acrosomal response [15,16]. Therefore, in this context, one could fear that depression treatment using SSRIs may worsen the already compromised fertility of depressive male patients.

At the molecular level, a common trait between male infertility and depression disorder is the observation that these situations are associated with the activation of pro-inflammatory pathways, as evidenced by the induction of oxidative stress, apoptosis and their consequences in terms of cell and nuclear damage [17–19]. One major contributor to these adverse responses is assumed to be associated with the dysfunction of the one-carbon cycle (1CC), one of the most important cycles for cell homeostasis centered on folate and vitamin B cycles providing methyl groups for the synthesis of DNA, polyamines, amino acids, creatine, phospholipids and antioxidants [20,21]. There are several studies to demonstrate that absence or deficiency in any of the central elements of the 1CC including folate, vitamin B, methionine, cysteine and homocysteine ends-up in impaired spermatogenesis, low sperm quality, and reduced fertility [22–25]. Very recently, it was reported that half of couple's referring to infertility center have MTFHR mutation (s) with hyper-homocysteinemia [26]. Similarly, in depressed individuals, it was reported that alteration of the 1CC is associated with the etiology of the disease [27,28]. In addition, a large body of literature presents folate deficiency, vitamin B12 deficiency, hyper-homocysteinemia and mutations in methylenetetrahydrofolate reductase (MTHFR), a key regulatory enzyme in folate and homocysteine metabolism, as risk factors for depression [29].

In this study, we aimed to show the effect of the antidepressant paroxetine, a serotonin reuptake inhibitor, on spermatogenesis and sperm function using an animal model inducing depression by the forced swim test (FST) [30,31]. In addition, we evaluated the major components of the one-carbon cycle such as folate, homocysteine, methionine, serine, glycine and vitamin B12 in this condition.

## Materials & methods

This study was approved by the ethics committee of the Royan Institute (IR.ACECR. ROYAN.REC.1399.084). All animal protocols and experiments were performed at the Royan Institute of Animal Biotechnology (Isfahan, Iran) in accordance with animal welfare guidelines. Mice were maintained under standard conditions: 12-hour light/12-hour dark cycle, temperature of 18-23˚C (65-75˚F) and humidity of 40-60%. Food and water were provided *ad libitum*.

For the current study, thirty mature male NMRI mice (6-8 weeks of age, 30±5 g) were used. As shown in Fig 1, the mice were divided into five groups (n= 6 in each group): a control group (group C), a sham group receiving only a saline solution (group Sa), a group treated with paroxetine at 7 mg/kg (group Par), a group depressed as a result of the stress applied as described below (group S), and finally, a depressed group treated with paroxetine (group SP). Animals were administered paroxetine via gavage with a daily dose of 7mg/kg paroxetine for 35 days, a concentration that founds its rationale following current treatments given in human [32] with the use of the human-to-animal dose conversion formula [33]. Following a pilot assay in which we treated NMRI mice with 3, 5, 7 or 9 mg/kg of paroxetine in order to find out what was the minimal paroxetine concentration affecting sperm structure and function. Therefore, we used a daily dose of 7mg/kg paroxetine in this study.

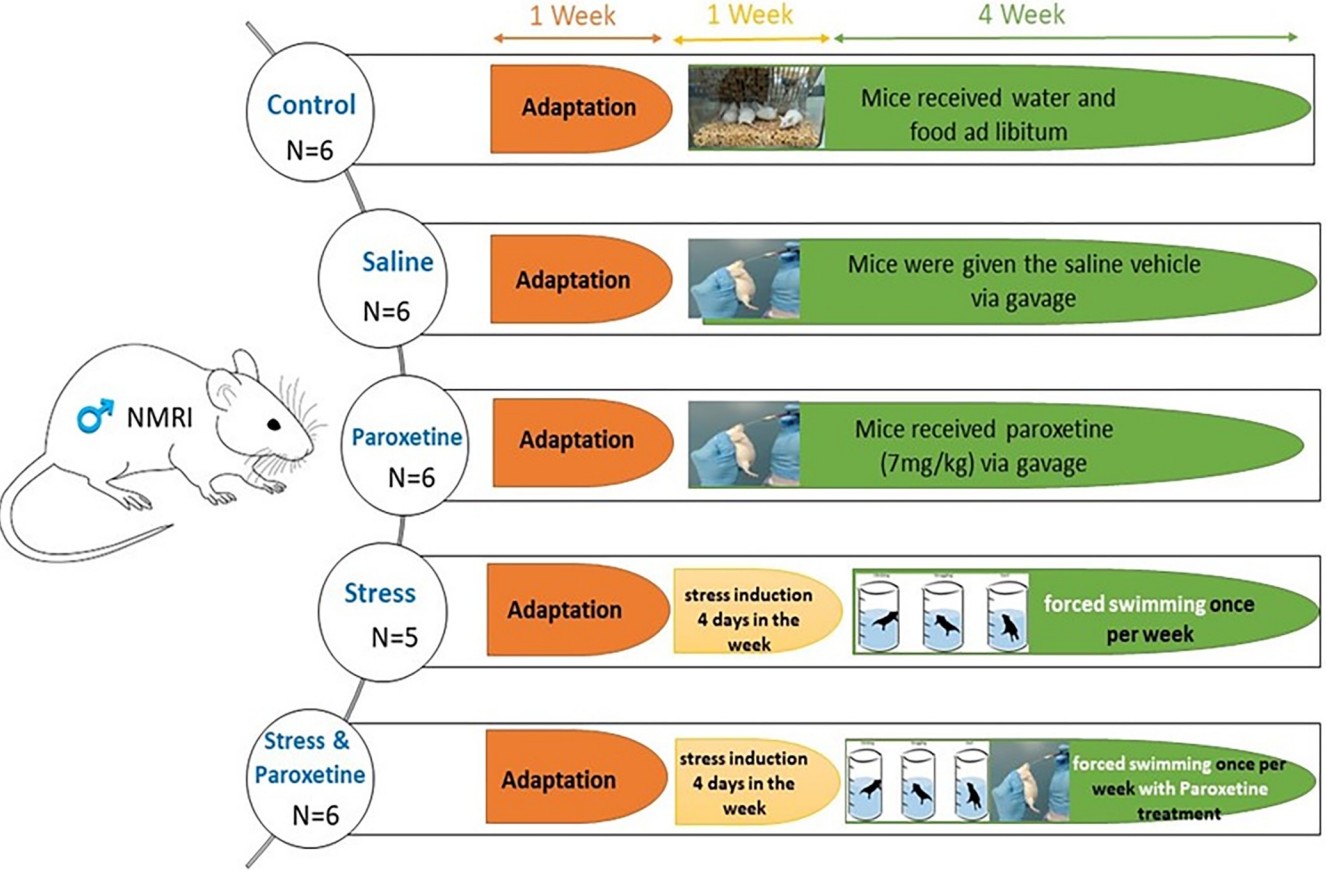

**Fig 1. Flowchart of study design in animal model.**

## Forced swim test

Like a depressed person who no longer tries to escape from unpleasant living conditions when previous attempts have failed, an animal exposed to constant stress gradually loses all hope of escaping and becomes immobile. The forced swim test (FST) mimics this situation and is a model for the induction of a depressed state in rodents. Unable to escape from the bathtub in which we placed them, the animals have diminished active behavioral responses and opt for immobility. Our apparatus consisted of immersing the animals in a bathtub 13-15cm deep in water at 32-34˚C for 15 min. The first week this test was performed for 4 consecutive days and repeated once for the next 4 weeks. At the end of the fourth swim, the behavior of the mice was recorded by video and the duration of the swim and the immobility of the animals were evaluated. Antidepressant treatment in the relevant groups was initiated 24 hours after the fourth swim. The recorded videos of the animals' behavior were evaluated to determine the extent of depression in each group and the efficacy of the antidepressants [34–36].

## Histological assessments

After 35 days, each mouse was weighed before euthanasia. Immediately, the testes, seminal vesicles, epididymides, kidneys, and liver were dissected and weighed individually. One testis from each animal was fixed in formalin for hematoxylin/eosin staining of paraffin sections (5-6 μm thickness) and evaluation of testicular parameters, including Johnsen score, tubular differentiation index (TDI), and spermatogenic index (SSI) as described earlier. For the TDI and SPI assessments, a minimum of 200 sections of seminiferous tubules were observed at 40× magnification using a light microscope (CX31 OLYMPUS,). For the Johnsen score examination, 100 seminiferous tubules were selected and scored on a scale of 1 to 10 at 40× magnification using a light microscope (CX31 OLYMPUS) [37].

## DNA fragmentation and apoptosis

The apoptosis was assessed by TUNEL kit based on manufacturer instructions. Following the de-paraffinized and rehydrated of 5–6 μm histological sections, they were treated for 30 min with 1 μl proteinase K (15 μg/ml in 10 mM Tris/Hcl, pH: 7.4) added to the slides. Then, the slides were washed with Phosphate-buffered saline solution (PBS). Subsequently, the slides were treated with 25 μl TUNEL solution (50 μl of enzyme solution and 450 μl label solution, 60 min). Then, they were covered by 25 μl POD-convertor (30 min) and subsequently incubated with 25 μl DAB substrate (1 μl for DAB and 10 μl of DAB substyrate, 60 sec). Finally, sections were counterstained with hematoxylin. Then, the apoptotic germ cells number/seminiferous tubule with same criteria were counted and compared between groups [38].

## Caspase-3 assay

The de-paraffinized and rehydrated slides were used for the antigen retrieval (using 10 mM sodium citrate buffer (pH: 7.2). After peroxidase blocking stage, the slides were washed with PBS and incubated at 4˚C for 18 hours with Caspase-3 primary antibody in humidified condition. Then, the slides were treated for 30 minutes with the secondary antibody, followed by streptavidin–HRP for 20 minutes at room temperature. Finally, the slides were treated with chromogen and counterstained with hematoxylin for10 seconds. Similar to TUNEL staining, the caspase-3+ germ cells number/seminiferous tubule with same criteria were counted and compared between groups [39,40].

## Evaluation of FSH, LH, and major components of 1CC

Blood plasma from the animals was stored at -20˚C for evaluation of hormones (FSH, LH) and major components of 1CC, including folate, vitamin B12, methionine, and homocysteine. In addition, glycine and serine contents were assessed. FSH, LH, folate, and vitamin B12 were quantified using radioimmunoassay (RIA) kits (DiaSorin, Italy) while the other molecules were quantified by high-performance liquid chromatography (HPLC) with an Agilent 1100 series, Dionex P680 system (Perkin Elmer serie 200, LC Packings nHPLC, Siemens,USA). In this study, all the plasma tests were carried out on 3 mice for each group.

## Sperm preparation and sperm parameter assessments

Spermatozoa were collected after mincing cauda epididymides in 1ml washing medium (VitaSperm™ Innovative Biotech, Tehran, Iran) at 37˚C for 30 min. To assess sperm concentration a sperm counting chamber was used (Sperm meter, Sperm processor, India) and data were expressed in million cells *per* ml. Sperm motility was evaluated on a pre-warmed slide and at least 200 cells were monitored. Sperm morphology was evaluated using eosin-nigrosine staining as described earlier [41]. Different types of aberrant spermatozoa morphologies (head, neck and tail regions) were monitored and expressed in percentage of the total number of sperm cells present in the sample.

For sperm membranous lipid peroxidation, the BODIPYⓇ 581/591 C11 test was used. Briefly, two million washed spermatozoa were exposed to the BODIPY C11 probe at a final concentration of 5 mM for 30 minutes at 37˚C. A positive control was performed for each sample by adding $H_2O_2$ to the sample. The samples were then washed twice (500g-5 min) with PBS 1X to remove the unbound BODIPY C11 probe. Finally, a flow cytometer FACS Calibur (Becton Dickinson, San Jose, CA, USA) was used to assess lipid peroxidation. For each sample, the percentage of BODIPY positive spermatozoa was recorded [41].

Sperm nuclear protamination was assessed using chromomycin A3 (CMA3) staining. Briefly, 20 μl of Carnoy fixative (methanol/acetic acid, 3: 1) was added to 20 μl of washed semen sample for 5 minutes. Next, a smear was prepared and the slides were air-dried at room temperature before treatment with CMA3 staining in McIlvaine buffer (7 ml 0.1 M citric acid + 32.9 ml $Na_2HPO_4$ .$7H_2O$, 0.2 M, pH 7.0, containing 10 mM $MgCl_2$) for 1 hour. After washing with phosphate-buffered saline (PBS 1X), the slides were mounted, and 200 sperm cells in each slide were evaluated using a fluorescent microscope (Olympus, BX51, Tokyo, Japan) with a 460 nm filter. Dark yellow spermatozoa (CMA3$^-$) were considered normally protaminated while light yellow cells (CMA3$^+$) were considered protamine-deficient spermatozoa [41].

To complete our evaluation of the proper organization of the sperm nucleus, we also assessed the nuclear histone content using aniline blue (AB) staining as reported earlier [42]. In this assay, immature spermatozoa (AB$^+$ sperm cells) turn dark blue due to high abnormal persistent histone content while normally mature sperm cells appear light blue.

In fine, sperm DNA integrity was also assessed using acridine orange (AO) staining as previously described. In this test, sperm with DNA damage (primarily DNA fragmentation) appear orange/red while sperm with good DNA integrity appear green. In each of these tests (sperm nuclear protamine content, sperm nuclear histone content, sperm DNA integrity), the results were expressed as a percentage [41].

## Statistical analysis

Verification of data normality was performed using the Shapiro-Wilk test before statistical analysis. Results were expressed as mean ± standard error, and significance between study groups was determined by Tukey's post hoc multiple comparison test using SPSS software. P

values less than 0.05 were considered statistically significant. All graphs were plotted using Graph Pad Prism.

## Results

### Animal weight, organ weights and morphometric parameters of testis and epididymis in the different animal groups

Body weight, kidney weight, seminal vesicle weight, testicular weight, epididymal length, and testicular volume were compared between groups. No significant differences were observed between groups. Only liver weight was found to be significantly higher in the paroxetine-treated group of depressed animals (2 g ± 0.7) compared with the control (2 g ± 0.1) and (saline) sham (2 g ± 0.1) groups (p<0.05). Fig 2 shows representative photographs of testis sections stained with H&E (upper panels) that were used to assess Johnsen score, SPI, and TDI (lower graphs). Mean percentages of Johnsen scores were not statistically different between

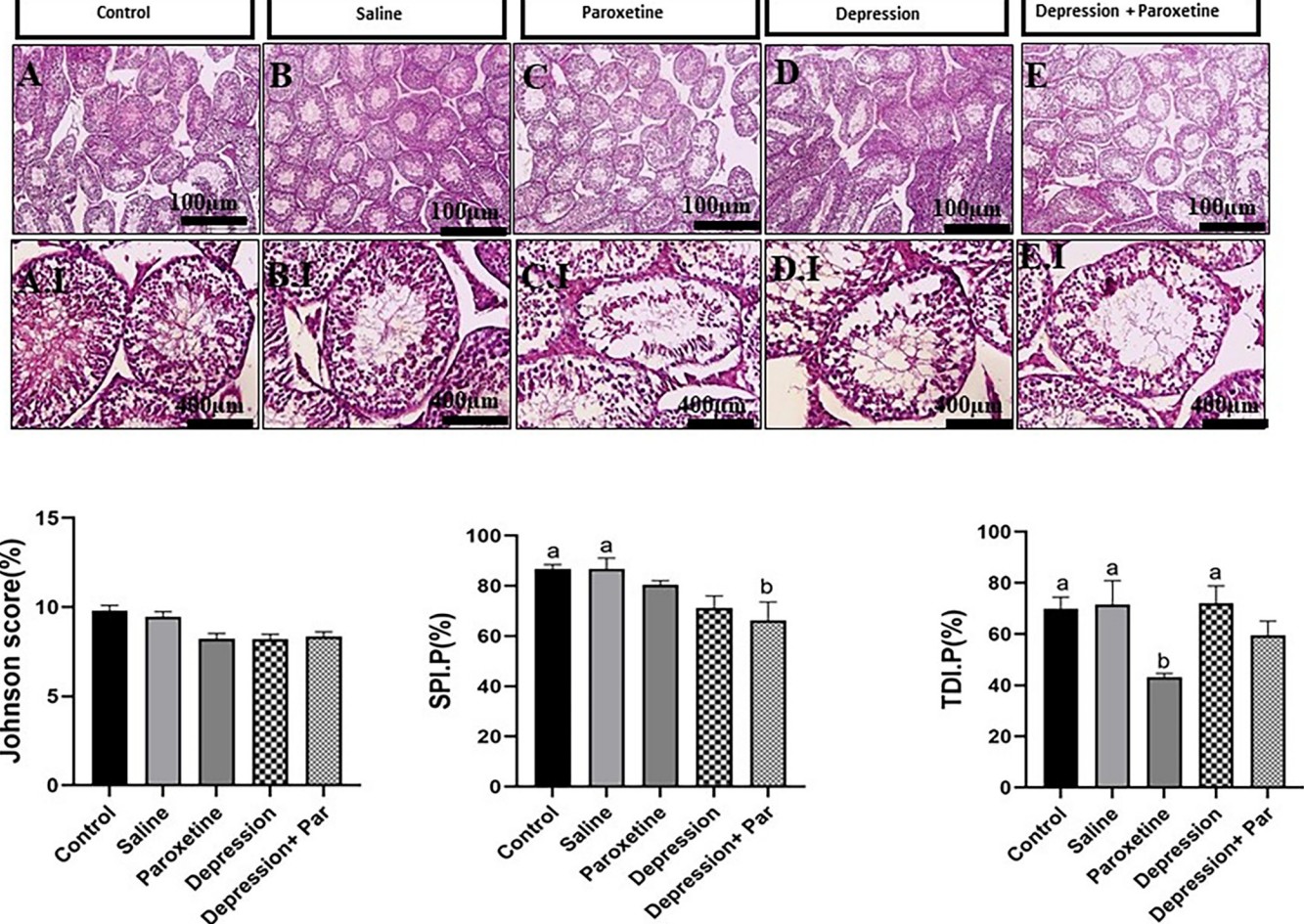

**Fig 2. Histological illustration of mouse testicular tissues.** *Top panels*: Comparison of testis sections between groups [A: Control, B: Saline (sham), C: Paroxetine (7 mg/kg), D: Stress (FST), E: Stress + Paroxetine (7 mg/kg)] at two different magnifications (100μm: Upper photographs; 400μm: Lower photographs). *Bottom bar graphs*: Comparison of testicular morphometric parameters including percentages of Johnsen score (left), the tubular differentiation index = TDI (middle), and spermatogenic index = SPI (right) between groups. Bars with different superscript letters are significantly different (p<0.05). Par=paroxetine.

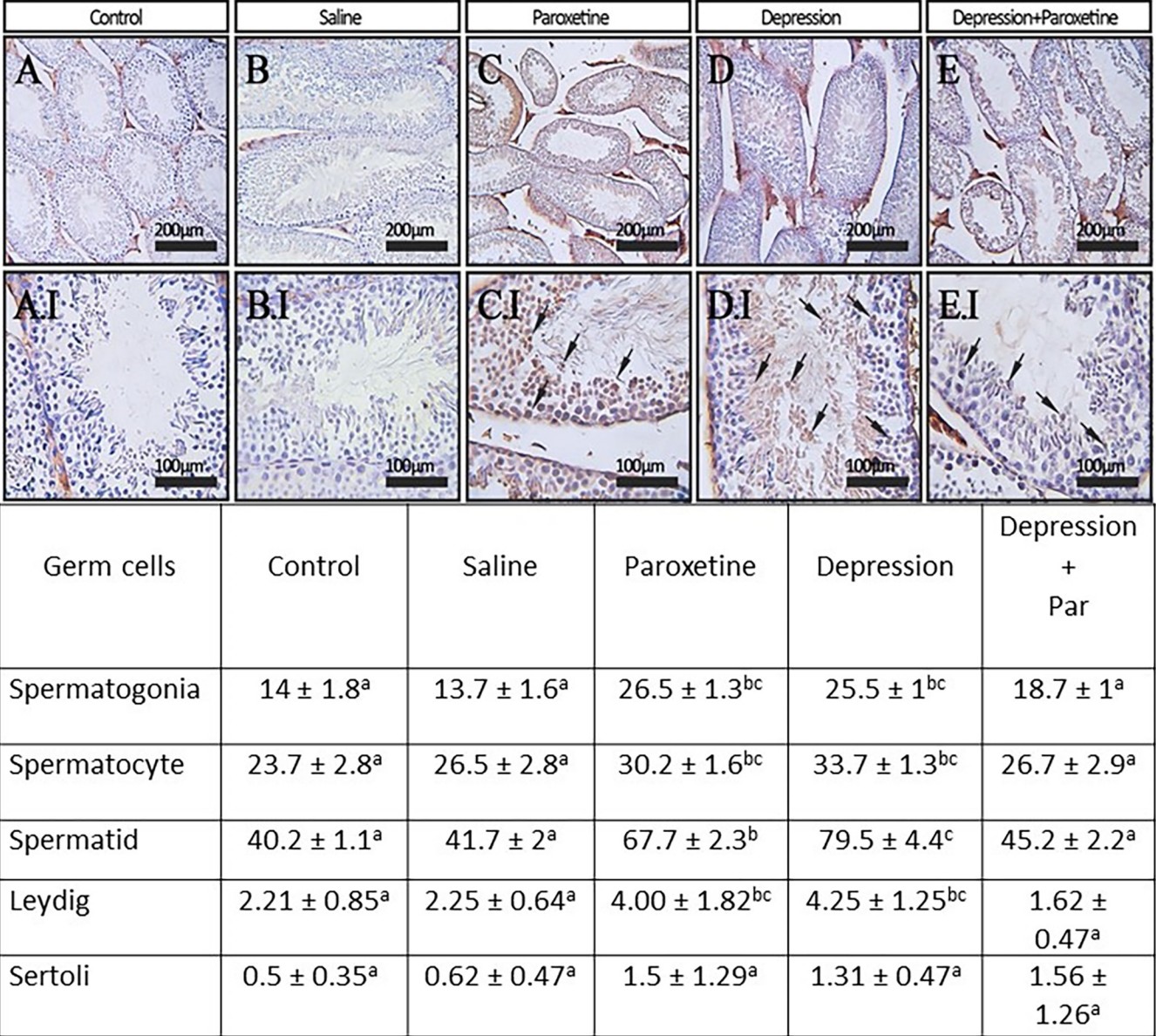

**Fig 3. Immunohistochemical staining of CASPASE-3 in testes cross-sections.** Comparison of mean percentage of Caspase-3 -positive cells in germ cells (spermatogonia, spermatocytes, spermatids, Leydig, and Sertoli cells) within groups. The values with different superscript letters in each row are significantly different (p≤0.05).

groups. The TDI was found to be significantly lower in the paroxetine group compared to the other groups (p<0.05), while the SPI was found to be significantly reduced only in the depressed animals treated with paroxetine compared to the control and sham (saline) groups (p<0.05).

In this study, two main apoptotic markers; TUNEL and caspase-3 were assessed on testicular tissue sections. As shown in Figs 3 and 4, strongest apoptotic damages were in paroxetine and depression groups compared to the control, saline, and depression plus paroxetine groups.

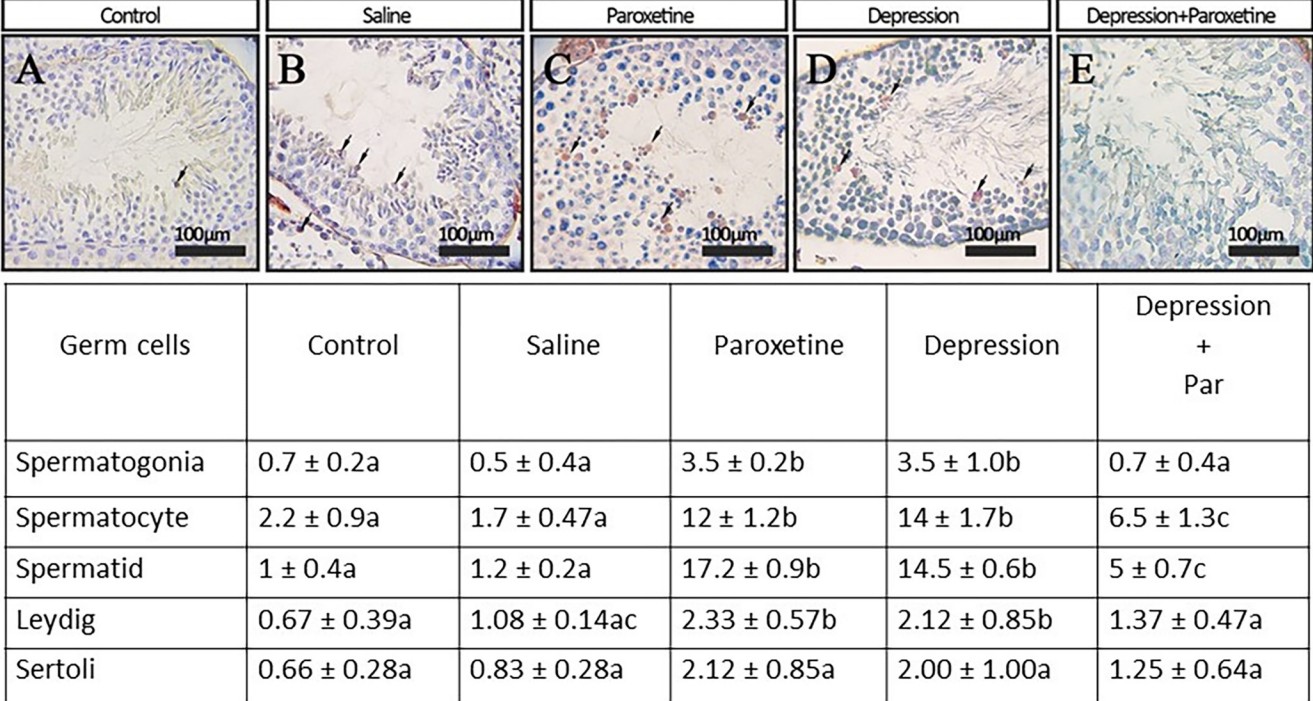

| Germ cells | Control | Saline | Paroxetine | Depression | Depression + Par |
|---|---|---|---|---|---|
| Spermatogonia | 0.7 ± 0.2a | 0.5 ± 0.4a | 3.5 ± 0.2b | 3.5 ± 1.0b | 0.7 ± 0.4a |
| Spermatocyte | 2.2 ± 0.9a | 1.7 ± 0.47a | 12 ± 1.2b | 14 ± 1.7b | 6.5 ± 1.3c |
| Spermatid | 1 ± 0.4a | 1.2 ± 0.2a | 17.2 ± 0.9b | 14.5 ± 0.6b | 5 ± 0.7c |
| Leydig | 0.67 ± 0.39a | 1.08 ± 0.14ac | 2.33 ± 0.57b | 2.12 ± 0.85b | 1.37 ± 0.47a |
| Sertoli | 0.66 ± 0.28a | 0.83 ± 0.28a | 2.12 ± 0.85a | 2.00 ± 1.00a | 1.25 ± 0.64a |

**Fig 4. Immunohistochemical staining of TUNEL positive cells in testes cross-sections.** Comparison of mean percentage of TUNEL-positive cells (spermatogonia, spermatocytes, spermatids, Leydig, and Sertoli cells) within groups. The values with different superscript letters in each row are significantly different (p≤0.05).

### Sperm parameters and sperm functional tests in the different animal groups

Mean sperm concentration (million/ml) and total motility were significantly decreased in the paroxetine and depression groups compared with the control and sham (saline) groups (Figs 3B–5A). In addition, the mean percentages of spermatozoa with abnormal morphology, increased histone content, decreased protamine content, and higher levels of lipid peroxidation (see Fig 3F–5C, respectively) were found to be higher in these same paroxetine and depression groups compared to the other groups (p<0.05). Regarding sperm DNA damage, only the paroxetine group had a significantly higher value (Fig 5G).

### Evaluation of plasma FSH and LH and 1CC components in the different animal groups

FSH and LH levels were significantly higher in the paroxetine and depression + paroxetine groups compared with the other groups (see Fig 6). With regard to the components of the carbon cycle (1CC), the level of folate was significantly higher (p<0.05) in the depression + paroxetine group only (see Fig 7) compared with the other groups. Homocysteine level was found to be significantly higher only in the paroxetine group (p<0.05). In contrast to methionine and serine levels, which were similar between groups, glycine levels were significantly lower in the paroxetine and depression plus paroxetine groups (see Fig 7). Similarly, vitamin B12 levels were significantly lower in the paroxetine and depression plus paroxetine groups compared with the other groups (see Fig 7).

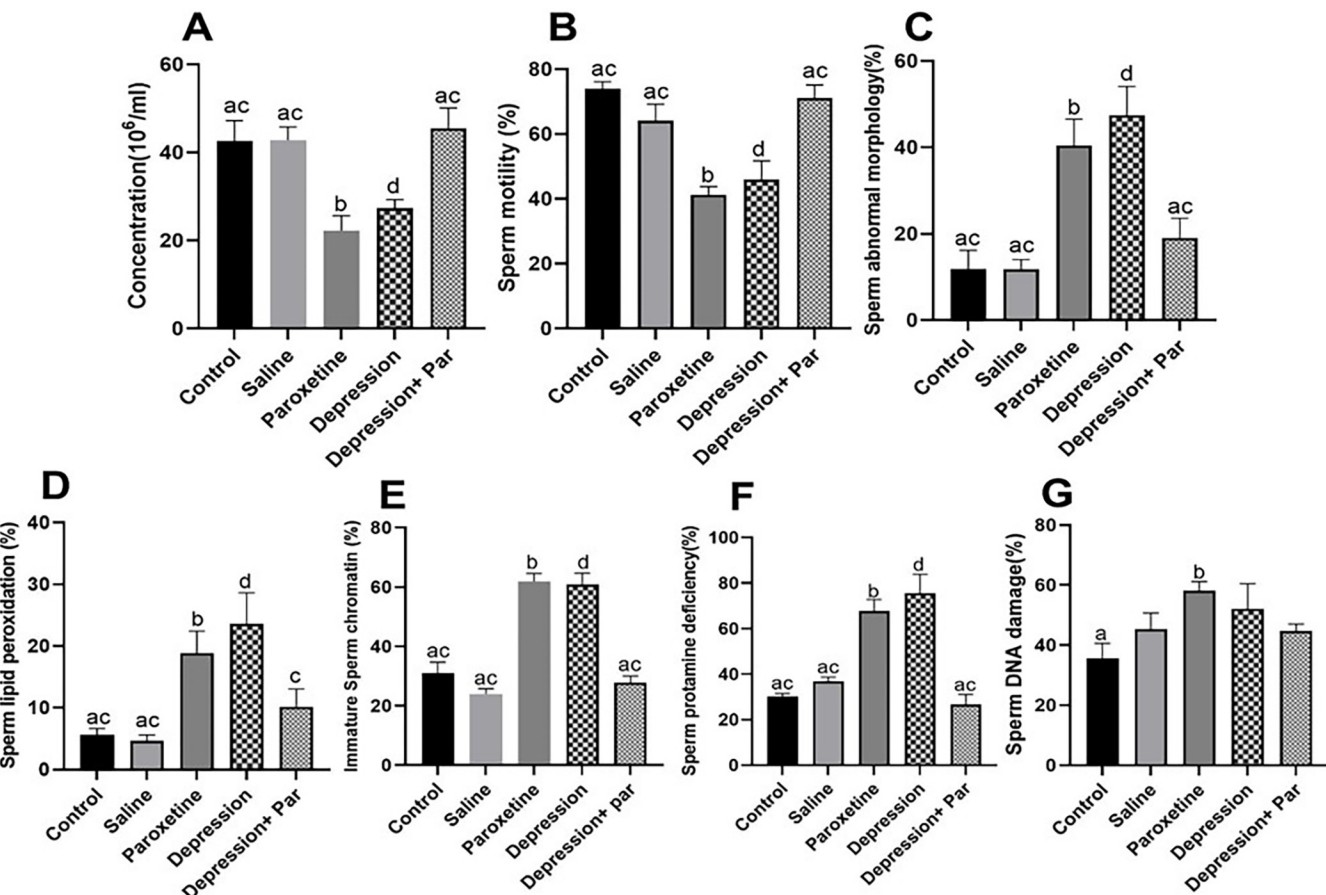

**Fig 5. Comparison of mouse sperm parameters in the different groups (N=6 for each group). A**: Concentration (in million /ml). **B**: Motile spermatozoa (in %). **C:** Spermatozoa with abnormal morphology (in %). **D:** Spermatozoa showing lipid peroxidation (in %). **E:** Spermatozoa having immature sperm chromatin (in %). **F:** Spermatozoa showing protamine deficiency (in %). **G:** Spermatozoa showing DNA damage (in %). Bars with different superscript letters are significantly different (p<0.05). Par = paroxetine.

## Discussion

In a global context where depressive states, as well as infertility, are on the rise, appropriate management of both conditions is necessary. Whether infertility is the cause or consequence of depression is a difficult question to answer, as it is clear that a significant portion of couples who seek infertility treatment also suffer from depression. This is nevertheless a controversial topic because for some authors, depression and anxiety arise from infertility [14] while for others, infertility is one of the consequences of depression [14,43,44]. As is often the case in these situations, the truth probably lies in the middle. Nesse et al suggested that in depressed states, fertility is not a priority function and that it is not surprising to expect its impairment, most likely through central nervous system imbalance and its roles on gonadal functions [45]. In this context, the question of whether antidepressants should be given to infertile patients has received some attention.

Using a forced swim test (FST) to generate a depressed state, we show here that depression does have deleterious effects on spermatogenesis and sperm function in this animal model. This was evidenced by the fact that in the group of animals under stress, sperm concentration and motility were significantly decreased. In addition, sperm quality also decreased, as evidenced by increased sperm nuclear immaturity (lower protamine content and higher histone

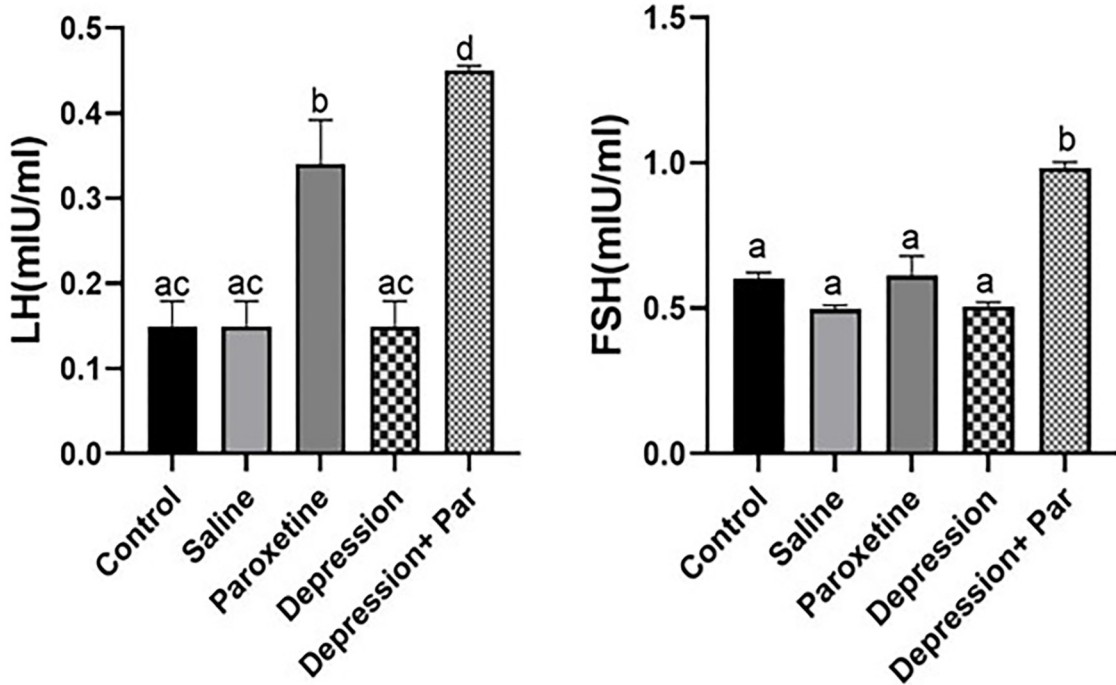

**Fig 6. Comparison of serum follicle-stimulating hormone (FSH) and Luteinizing hormone (LH) levels in mouse serum between groups (N=3 for each group).** Bars with different superscript letters are significantly different (p<0.05). Par = paroxetine.

retention). Intriguingly, these negative impacts on sperm production, integrity, and function were not associated with significant changes in FSH and LH, nor with changes in the components of the one-carbon cycle as we hypothesized initially. It remains to be determined what mechanisms are at play to explain the effects of FST. Although most studies have introduced FST model to induction of depression in animals [34–36], it is possible that this model leads to the induction of stress more than depression in animals. Or the duration of this model can only create acute conditions that can be completely different from chronic depressive conditions in humans. However, more studies are needed in this regard.

To the question of whether antidepressants can impair male fertility, we show here that paroxetine, one of the SSRIs commonly used to treat depressed patients, when administered to healthy animals, affects sperm production and quality. First, paroxetine-treated animals showed an increase in serum FSH and LH levels. This suggests that in healthy mice, paroxetine altered PHA. Surprisingly, in a setting where paroxetine stimulates FSH and LH secretion, we do not register the expected improvement in sperm production. Unlike our study, Yardimci et al assessed the effects of long-term paroxetine treatment on puberty onset in adolescent male rats. They reported that treatment with paroxetine did not result in any changes in the level of FSH, LH and leptin [46]. The reason for the difference between our results and others requires further investigation.

In addition, we observed that the mean of tubular differentiation index, and apoptotic markers such as CASPASE-3-and TUNEL-positive cells in testicular germ cells significantly higher in the paroxetine-treated group of animals compared to control group. This suggests that in our condition, despite PHA stimulation of LH, paroxetine impairs spermatogenesis by another mechanism. Second, paroxetine also alters sperm structure and function since motility was decreased and the percentage of abnormal sperm was increased. In addition, paroxetine

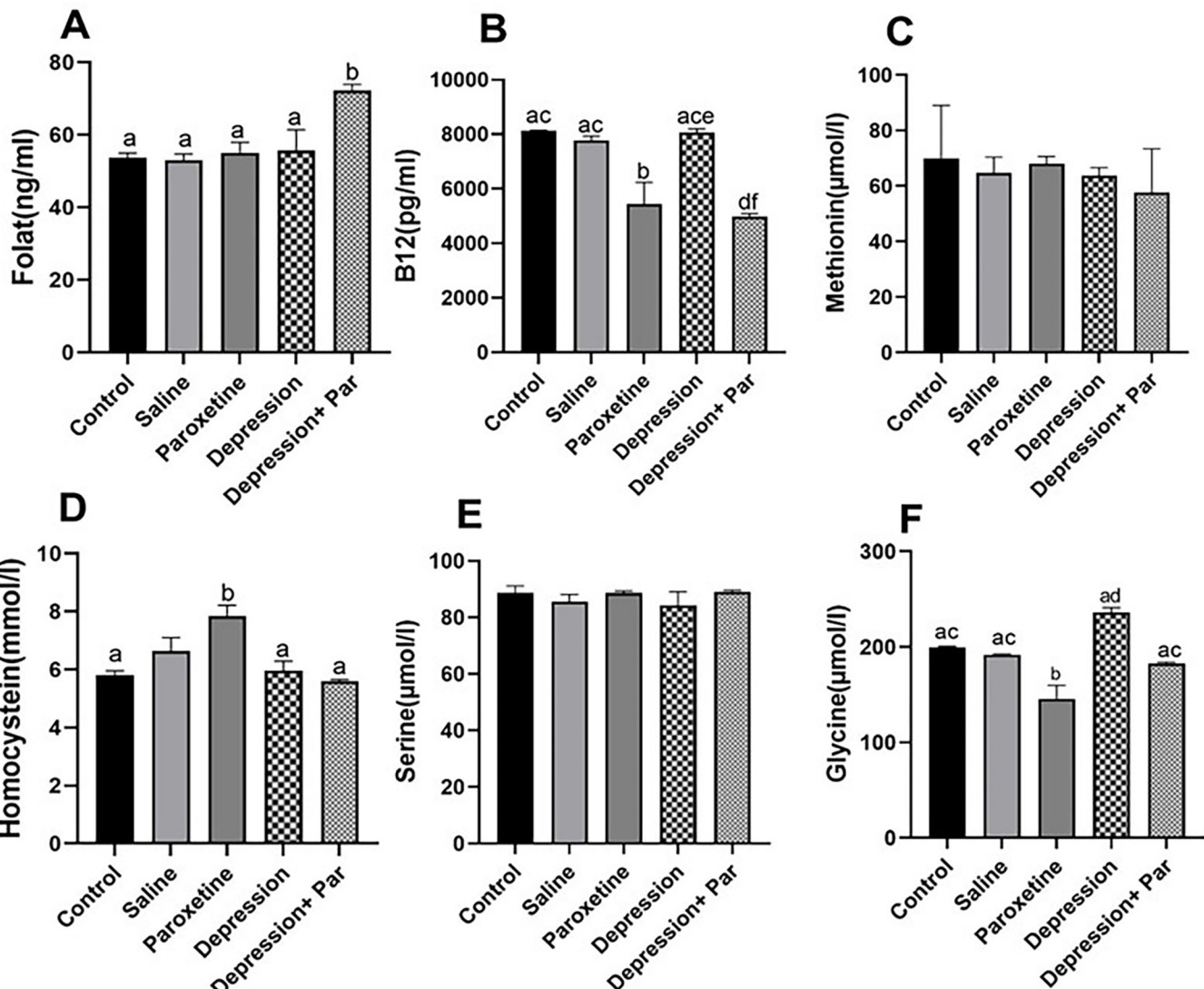

**Fig 7. Evaluation of one-carbon cycle indicators between mouse groups (N=3 for each group).** Bars with different superscript letters are significantly different ($p<0.05$).

treatment resulted in increased sperm membrane lipid peroxidation, as well as nuclear immaturity (evidenced by a lower level of protamination and a higher level of persistent histone) and sperm DNA damage. Therefore, by itself, paroxetine has reprotoxic effects when used in healthy animals. These observations are in agreement with previous studies showing that paroxetine induces mitochondrial damage, ROS production, and cell apoptosis [9,47].

A few studies have examined the effect of SSRI antidepressants on sperm parameters and fertility potential in depressed individuals [9,11,15]. It was generally concluded that most of the tested medications did not have a severe negative impact on spermatogenesis and fertility [9,48]. These data are difficult to compare with ours because in addition to the fact that these were human studies, the duration of treatment was also quite different and corresponded more to chronic exposure than to acute exposure as in our case. However, SSRI antidepressants have not been completely free of some negative effects on male fertility [15,43,49,50]. Specifically, paroxetine has been shown to be associated with negative side effects, one of

which is impaired gametogenesis and fertility, a point we confirm here in our animal model. It has been hypothesized that this was primarily due to the action of paroxetine on the pituitary-hypothalamic axis (PHA), with serotonin being a neurotransmitter orchestrating the secretion and release of gonadotropins [5,6]. This hypothesis does not appear to be supported by our current data.

Intriguingly, we show here that when paroxetine is administered to depressed animals, it no longer had an adverse effect on sperm production and quality. On the contrary, paroxetine treatment of depressed animals was associated with an even greater increase in serum FSH and LH levels than when administered to healthy animals. This was accompanied by a restoration of sperm concentration although we did not record a significant improvement in testicular histology since neither TDI nor SPI were improved. We also observed that the mean of apoptotic markers (CASPASE-3-and TUNEL-positive cells) in testicular germ cells did not significantly increase in the depressed animals that were treated with paroxetine compared to other groups, and was similar to control and saline groups. In addition, sperm motility returned to levels seen in control animals. Only sperm morphology was not normalized. In addition, the 4 measured parameters giving indications of sperm maturity and damage (membrane lipid peroxidation, nuclear histone and protamine content, DNA damage) were normalized to levels close to controls, attesting to the beneficial effect of paroxetine treatment on the male reproductive axis of depressed animals.

To better understand what is going on and to test our initial hypothesis that depression and/or antidepressants might alter the one-carbon cycle, which is so important for cellular homeostasis and in particular for the management of oxidative stress, we monitored key components of the one-carbon cycle in the different groups of animals. We show that paroxetine, but not depression in the animal model we used, resulted in systemic hyper-homocysteinemia which may explain the oxidative impairment observed on reproductive function in healthy paroxetine-treated animals. Consistent with the hyper-homocysteinemia, paroxetine-treated animals also had lower levels of vitamin B12, which is essential for converting homocysteine to methionine [51] and, in addition, had a lower serum content of glycine, another important player in homocysteine metabolism [52]. However, despite higher homocysteine levels, the conversion of homocysteine to methionine was not altered to the extent that methionine levels were modified as all groups of animals followed had similar serum methionine levels. Under our conditions, therefore, it appears that paroxetine treatment, but not depression, impacts 1CC balance. This could likely generate systemic oxidative stress that could explain the adverse effects of paroxetine on the male reproductive axis that is so finely tuned by oxidative reactions [53,54]. Intriguingly again, and consistent with the data reported above, when paroxetine was administered to depressed animals, most of these changes in 1CC components returned to control levels. Only the serum vitamin B12 level remained low as it was in the paroxetine-treated animals. Interestingly, serum folate levels were significantly higher in the depressed paroxetine-treated animals. This point deserves some attention because folic acid is an important molecule for DNA methylation and genetic imprinting, which are key phenomena for optimal spermatogenesis [22]. Since testicular folate levels mirror serum folate levels, one would expect to have higher folate levels in the testes of depressed animals treated with paroxetine [55]. This could be beneficial in terms of fertility potential as folate supplementation has recently been shown to improve IVF-ISCI outcomes, an action that was assumed to be related to its strong antioxidant properties [56,57]. It should be noted that we evaluated the components of 1 CC at the serum level, and that additional studies of the expression of 1CC players conducted in the testis might be relevant.

In conclusion, using an animal model of induced depression, we show that depression is associated with adverse effects on sperm production and integrity. We also show that a

commonly prescribed antidepressant of the SSRI class (paroxetine) has male reproductive negative effects when used on healthy animals. Somewhat surprisingly, when given to depressed animals at the same dose, paroxetine treatment normalized all of the negative effects on male reproduction recorded in the depressed animals. Our initial hypothesis that an imbalance in the one-carbon cycle might be the common feature that would explain both the adverse effect of depression and paroxetine treatment on male reproductive fitness was not confirmed by our data, because depression (in the animal model used) did not appear to significantly affect the one-carbon cycle, whereas paroxetine did.

## Supporting information

**S1 Data.**
(XLSX)

## Acknowledgments

The authors would like to express their gratitude to the staff of the Animal Biotechnology Department of the Royan Institute for their full support. In addition, the authors thank Mohsen Rahmani for training Reyhane Aghajani with experimental work.

## Author Contributions

**Conceptualization:** Marziyeh Tavalaee, Parviz Gharagozloo, Joël R. Drevet, Mohammad Hossein Nasr-Esfahani.

**Data curation:** Reyhane Aghajani, Mohammad Hossein Nasr-Esfahani.

**Formal analysis:** Reyhane Aghajani, Marziyeh Tavalaee.

**Investigation:** Reyhane Aghajani, Niloofar Sadeghi, Mazdak Razi, Maryam Arbabian.

**Methodology:** Reyhane Aghajani, Niloofar Sadeghi.

**Supervision:** Marziyeh Tavalaee, Mohammad Hossein Nasr-Esfahani.

**Validation:** Mohammad Hossein Nasr-Esfahani.

**Writing – original draft:** Marziyeh Tavalaee, Joël R. Drevet, Mohammad Hossein Nasr-Esfahani.

**Writing – review & editing:** Marziyeh Tavalaee, Niloofar Sadeghi, Parviz Gharagozloo, Joël R. Drevet, Mohammad Hossein Nasr-Esfahani.

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
