## [Decision Letter · Decision Letter 0]

15 Nov 2021

PONE-D-21-29506Paroxetine treatment in an animal model of depression improves sperm qualityPLOS ONE

Dear Dr. Nasr-Esfahani,

Thank you for submitting your manuscript to PLOS ONE. After careful consideration, we feel that it has merit but does not fully meet PLOS ONE’s publication criteria as it currently stands. Therefore, we invite you to submit a revised version of the manuscript that addresses the points raised during the review process. The subject of the study is interesting. The presentation of the problem in the introduction and discussion is diffused. Analyzing additional parameters such as LH, FSH, Testosterone and makers for apoptosis are suggested.

We look forward to receiving your revised manuscript.

Kind regards,

Suresh Yenugu

Academic Editor

PLOS ONE

Journal Requirements:

Reviewers' comments:

Reviewer's Responses to Questions

**Comments to the Author**

1. Is the manuscript technically sound, and do the data support the conclusions?

Reviewer #1: Yes

Reviewer #2: Partly

2. Has the statistical analysis been performed appropriately and rigorously? 

Reviewer #1: Yes

Reviewer #2: Yes

3. Have the authors made all data underlying the findings in their manuscript fully available?

Reviewer #1: Yes

Reviewer #2: Yes

4. Is the manuscript presented in an intelligible fashion and written in standard English?

Reviewer #1: Yes

Reviewer #2: Yes

5. Review Comments to the Author

Reviewer #1: The manuscript entitled ‘Paroxetine treatment in an animal model of depression 1 improves sperm quality’ describes the effect of anti-depressent, paroxetine, in regulating sperm parameters in rodent depression model. They show that paroxetine exerts negative impacts on male reproductive function when administered to healthy animals; it perfectly corrects the altered sperm parameters of depressed animals.

1. Introduce the depression physiology like changes in the brain bit more in the introduction. Link it up with decreased fertility initially.

2. Is the correlation between depression and reduced fertility in males already studied?

3. How long the treatment for antidepressant was carried out in mice? Why that time frame was selected?

4. Authors should have included cortisol levels in all groups as reflection to stress. To me the protocol seems to be more as stress induction than depression though authors have taken reference of other workers using same model earlier.

5. Authors should have also included testosterone levels in all experimental groups?

6. Can they discuss why LH and FSH increased after anti-depressant treatment in depression induced mice?

7. Is it possible for them to also show the apoptosis markers in testes like TUNEL or caspase assay or gene expression in all groups?

Reviewer #2: The article entitled "Paroxetine treatment in an animal model of depression improves sperm quality" by the group of Prof. Mohammad Hossein Nasr-Esfahani is a timely one.

I have some general and specific queries:

1. The abstract should be more specific.

2. The introduction parts tells about many associations, but the plan of experiments are not accordingly.

3. Any evidence of the enhancement of one-carbon cycle (1CC) in treatment is desired, preferably at expression level.

4. All the figure legends should have the mention of the model animal.

5. The study does not indicate any specific pathway for the effect of the drug.

6. The discussion may be shorten. It is long as it has to suggest most possibilities about the pathway

6. PLOS authors have the option to publish the peer review history of their article (what does this mean?). If published, this will include your full peer review and any attached files.

Reviewer #1: No

Reviewer #2: **Yes: **Asamanja Chattoraj

---

## [Author Response · Author response to Decision Letter 0]

25 May 2022

PLOS ONE-D-21-29506

Paroxetine treatment in an animal model of depression improves sperm quality

Editor in Chief, PLOS ONE Journal

Dear Prof. Suresh Yenugu,

Thank you very much for giving us the opportunity to improve our manuscript entitled "Paroxetine treatment in an animal model of depression improves sperm quality". All suggestions made by the reviewers were extremely helpful and have been taken into account. Please find below point by point our responses to each comment/suggestion/request. 

The revised manuscript has been approved by all authors and the changes are highlighted in the document. We hope that the revised manuscript will live up to the expectations of the PLOS ONE editorial staff.

Dr. Mazdak Razi who performed the additional requested experiments has been added to the list of authors. 

On behalf of the authors.

Mohammad H Nasr-Esfahani & Joël R. Drevet

………………………………………………………………………………………………….

Editorial requests

The subject of the study is interesting. The presentation of the problem in the introduction and discussion is diffused. Analyzing additional parameters such as LH, FSH, Testosterone and makers for apoptosis are suggested.

Authors’ response: Thank you very much for your overall appreciative comment. 

We have slightly revised the introduction and discussion sections in hopes of providing more information about the purpose of this study and our proposed conclusions.

Per your suggestion and that of the reviewers, we have now evaluated apoptotic markers such as TUNEL and Caspase-3 in testicular tissue from all groups of animals and present these new data in the revised manuscript. 

This is the main reason for the slow return of our revised manuscript.

Unfortunately, no serum was available to assess LH, FSH and Testosterone levels. We mentioned in the discussion section that it would be relevant to do so in a follow-up study.

Authors’ response: To meet the editorial requirements of PLoS ONE, we have ensured that no reference to "data not shown" remains in the revised manuscript, either by providing additional information to find these data or an appropriate reference.

Reviewers' comments:

Reviewer #1: The manuscript entitled ‘Paroxetine treatment in an animal model of depression 1 improves sperm quality’ describes the effect of anti-depressent, paroxetine, in regulating sperm parameters in rodent depression model. They show that paroxetine exerts negative impacts on male reproductive function when administered to healthy animals; it perfectly corrects the altered sperm parameters of depressed animals.

1. Introduce the depression physiology like changes in the brain bit more in the introduction. Link it up with decreased fertility initially.

Authors’ response: At the reviewer's request, we have expanded the introduction section by providing general background regarding brain dysfunction in depressive states (Page 3, lines 4 to 7). In addition, we also provide more detail on how male reproductive function is thought to be impaired in depression (Page 3, lines 10 to 26).

2. Is the correlation between depression and reduced fertility in males already studied?

Authors’ response: As noted in our response to the previous point, we have provided more detail on the depression/reproductive male impact context in the Introduction section. We have also introduced more publications. 

3. How long the treatment for antidepressant was carried out in mice? Why that time frame was selected?

Authors’ response: we apologize for not clearly providing this information. We state in the revised manuscript that the animals were treated with a daily dose of 7mg/kg paroxetine for 35 days (page 5, lines 7 to 8). This duration was considered because it corresponds to one cycle of spermatogenesis in mice.

4. Authors should have included cortisol levels in all groups as reflection to stress. To me the protocol seems to be more as stress induction than depression though authors have taken reference of other workers using same model earlier. Authors should have also included testosterone levels in all experimental groups?

Authors’ response: The animal model of the forced swim test used in this study could indeed be considered a stress model, but when the animals repeatedly subjected to it give up the idea of escape and remain immobile, it is considered to induce a depressive situation. This model has already been validated as such. It is the most widely used animal behavioral test for antidepressant medications testing. It is also known as the Porsolt test, after its developer. For a representative reference among many, see (Yankelevitch-Yahav et al., 2015).

The reviewer and editor suggested that other markers could have been presented, such as testicular apoptotic markers and serum testosterone. Because we had preserved some testicular tissue from these animals, we were able to respond to the evaluation of apoptotic markers. Unfortunately, we no longer had sera from these sacrificed animals, which prevented us from providing the assessment of testosterone.

5. Can they discuss why LH and FSH increased after anti-depressant treatment in depression induced mice?

Authors’ response: We did observe that paroxetine-treated animals had increased serum FSH and LH levels. It has previously been suggested that paroxetine does alter the hypothalamic-pituitary-gonadal axis and stimulates FSH and LH secretion. However, a puzzling point is that we do not observe the expected improvement in sperm production that should logically follow. Differently, Yardimci et al (2018) evaluated the effects of long-term paroxetine treatment on the onset of puberty in adolescent male rats. They reported that paroxetine treatment resulted in no change in FSH, LH, and leptin levels in contrast to what we observed in mice. The reason for this discrepancy escapes us at this stage and requires further investigation. It could be related to the species (rat versus mouse), the age of the animals or the treatment conditions (dose and duration). We included this point in the revised manuscript.

6. Is it possible for them to also show the apoptosis markers in testes like TUNEL or caspase assay or gene expression in all groups?

Authors’ response: At the request of the reviewer, we now provide TUNEL and testicular Caspase3 evaluation in all animal groups (see figures 3 & 4).

Reviewer #2: 

The article entitled "Paroxetine treatment in an animal model of depression improves sperm quality" by the group of Prof. Mohammad Hossein Nasr-Esfahani is a timely one.

I have some general and specific queries:

1.The abstract should be more specific.

Authors’ response: at the request of the reviewer, an attempt has been made to revise the abstract in order to provide more specific details.

2. The introduction parts tells about many associations, but the plan of experiments are not accordingly.

Authors’ response: With all due respect, we are not sure we understand what the reviewer meant. Nevertheless, if we did, we have tried to better clarify the context of our study and the question(s) we asked by providing a short presentation at the end of the Introduction section. Figure 1 also describes in some detail the different situations we explored.

3. Any evidence of the enhancement of one-carbon cycle (1CC) in treatment is desired, preferably at expression level.

Authors’ response: Well, we agree with the reviewer. We show here that the plasma content of specific carbon cycle byproducts (i.e., glycine and vitamin B12) is reduced in the paroxetine and depression + paroxetine groups of animals compared to the other groups. Whether this is due to a genomic response or a non-genomic response is certainly an interesting point that will need to be investigated in the future. Because we did not study it, we mentioned in the revised discussion that this could be considered a limitation.

4. All the figure legends should have the mention of the model animal.

Authors’ response: at the reviewer request, the fact that the figures present data obtained using the mouse model has been indicated in the legend of each figure.

5. The study does not indicate any specific pathway for the effect of the drug.

Authors’ response: With all due respect, the supposed interference of SSRI antidepressants such as paroxetine is supposed to be with the HPG axis and this is already mentioned in the introduction section (page 3, lines 10 to 12).

6. The discussion may be shorten. It is long as it has to suggest most possibilities about the pathway. 

Authors’ response: we have shortened some subsections of the discussion, as suggested by the reviewer.

---

## [Decision Letter · Decision Letter 1]

27 Jun 2022

Paroxetine treatment in an animal model of depression improves sperm quality

PONE-D-21-29506R1

Dear Dr. Nasr-Esfahani,

We’re pleased to inform you that your manuscript has been judged scientifically suitable for publication and will be formally accepted for publication once it meets all outstanding technical requirements.

Kind regards,

Suresh Yenugu

Academic Editor

PLOS ONE

Additional Editor Comments (optional):

Reviewers' comments:

Reviewer's Responses to Questions

**Comments to the Author**

1. If the authors have adequately addressed your comments raised in a previous round of review and you feel that this manuscript is now acceptable for publication, you may indicate that here to bypass the “Comments to the Author” section, enter your conflict of interest statement in the “Confidential to Editor” section, and submit your "Accept" recommendation.

Reviewer #1: All comments have been addressed

Reviewer #2: All comments have been addressed

2. Is the manuscript technically sound, and do the data support the conclusions?

Reviewer #1: Yes

Reviewer #2: Yes

3. Has the statistical analysis been performed appropriately and rigorously? 

Reviewer #1: Yes

Reviewer #2: Yes

4. Have the authors made all data underlying the findings in their manuscript fully available?

Reviewer #1: Yes

Reviewer #2: Yes

5. Is the manuscript presented in an intelligible fashion and written in standard English?

Reviewer #1: Yes

Reviewer #2: Yes

6. Review Comments to the Author

Reviewer #1: The authors have replied to all the queries raised.

The manuscript now might be accepted from my side.

Reviewer #2: The authors addressed all the questions raised by me.

The manuscript can be accepted for the publication.

7. PLOS authors have the option to publish the peer review history of their article (what does this mean?). If published, this will include your full peer review and any attached files.

Reviewer #1: **Yes: **Arnab Banerjee

Reviewer #2: **Yes: **Prof Asamanja Chattoraj

---

## [Editor Report · Acceptance letter]

24 Nov 2022

PONE-D-21-29506R1 

Paroxetine treatment in an animal model of depression improves sperm quality 

Dear Dr. Nasr-Esfahani:

I'm pleased to inform you that your manuscript has been deemed suitable for publication in PLOS ONE. Congratulations! Your manuscript is now with our production department. 

Kind regards, 

on behalf of

Dr. Suresh Yenugu 

Academic Editor

PLOS ONE